# Phosphazene-Containing Epoxy Resins Based on Bisphenol F with Enhanced Heat Resistance and Mechanical Properties: Synthesis and Properties

**DOI:** 10.3390/polym14214547

**Published:** 2022-10-27

**Authors:** Ilya V. Tarasov, Anastasiya V. Oboishchikova, Roman S. Borisov, Vyacheslav V. Kireev, Igor S. Sirotin

**Affiliations:** 1Department of Plastics, Mendeleev University of Chemical Technology of Russia, Miusskaya sq. 9, 125047 Moscow, Russia; 2Topchiev Institute of Petrochemical Synthesis, Russian Academy of Sciences, Leninskii pr. 29, 119991 Moscow, Russia

**Keywords:** epoxy resin, epoxy oligomer, phosphazene, bisphenol F, mechanical properties, glass transition temperature

## Abstract

Organophosphazenes are of interest due to the combination of increased mechanical and thermal properties of polymer materials obtained with their use, however, they are characterized by a complex multi-stage synthesis. Moreover, the high viscosity of phosphazene-containing epoxy resins (PhER) makes their processing difficult. To simplify the synthesis of PhER, a one-step method was developed, and bisphenol F was chosen, which also provided a decrease in viscosity. In the current study, PhER were formed by a one-stage interaction of hexachlorocyclotriphosphazene (HCP) with bisphenol F isomers and epichlorohydrin in the presence of alkali, which was a mixture of epoxycyclophosphazenes (ECPh) with a functionality from 1 to 4 according to the results of MALDI-TOF analysis. Conventional epoxy resins based on bisphenol F, also formed during the process, showed high mechanical properties and glass transition temperature, and the reactivity of the obtained resins is similar to the base epoxy resins based on bisphenols A and F. Cured PhER had higher or the same mechanical properties compared to base epoxy resins based on bisphenol A and F, and a glass transition temperature comparable to base epoxy resins based on bisphenol F: glass transition temperature (Tg) up to 174.5 °C, tensile strength up to 74.5 MPa, tensile modulus up to 2050 MPa, tensile elongation at break up to 6.22%, flexural strength up to 146.6 MPa, flexural modulus up to 3630 MPa, flexural elongation at break up to 9.15%, and Izod impact strength up to 4.01 kJ/m^2^. Analysis of the composition of the obtained PhER was carried out by ^1^H and ^31^P NMR spectroscopy, MALDI-TOF mass spectrometry, X-ray fluorescence elemental analysis, and contained up to 3.9% phosphorus and from 1.3% to 4.2% chlorine. The temperature profile of the viscosity of the resulting epoxy resins was determined, and the viscosity at 25 °C ranged from 20,000 to 450,000 Pa·s, depending on the ratio of reagents. The resins studied in this work can be cured with conventional curing agents and, with a low content of the phosphazene fraction, can act as modifiers for traditional epoxy resins, being compatible with them, to increase impact strength and elasticity while maintaining the rest of the main mechanical and processing properties, and can be used as a resin component for composite materials, adhesives, and paints.

## 1. Introduction

Epoxy resins are one of the most common thermosetting polymers, having a wide range of applications, such as: coatings, adhesives, casting compounds, resins, and binders for composite materials in various areas, from household to critical in aircraft, rocket, mechanical engineering, and electronics and electrical engineering [1].

At the same time, a number of thermoplastics and other types of thermosetting polymers are used for the same applications. A new generation of binders is being developed with enhanced mechanical properties, thermal and fire resistance, such as: bismaleimides, polyimides, and cyanate ethers [2]. However, they are inferior to epoxides in cost, manufacturability, and in some cases in mechanical properties and toxicity during processing [2,3].

With their advantages, base epoxy resins also have a number of disadvantages, as they often have low and unstable performance characteristics, in particular, combustibility and relatively low heat resistance [4]. An effective way to increase the heat resistance of epoxy matrices is to modify them with polyfunctional resins that crosslink and form a joint mesh structure with base resins. Examples of such resins are triglycidyl-p-aminophenol, tetraglycidyl-4,4′-methylenedianiline, and epoxidized novolaks [4,5].

The problem of flammability of epoxy resins is more complex and can be solved using various additives and/or components:Halogenated epoxy resinsAdditive-type fire retardantHalogen-free organoelement compounds

The traditional way to reduce flammability is to use halogen-containing epoxy resins, in particular based on brominated bisphenol A. However, these types of resins release toxic gases when in contact with a flame, which limits their use in areas that involve human contact [6,7].

The modern global trend is the use of halogen-free polymer materials as additives to improve the fire resistance of polymeric materials [6]. A cheap and common option is the use of inorganic additives, such as aluminum or magnesium hydroxides, polyphosphates, and red phosphorus, but they significantly reduce the mechanical properties and transparency of materials and degrade the technological properties important for processing [6,8].

Organophosphates proved to be more functional. They are better combined with the base polymer, but at the same time they can act as a plasticizer and reduce heat resistance [9].

Functional organoelement compounds combine the advantages of non-toxicity, preservation of the properties of the base polymer, and heat resistance, capable of forming a common three-dimensional structure with the base epoxy resin [7,10]. An example of such modifiers are glycidyl esters of phosphorus acids, with a decrease in flammability, which increase the mechanical and adhesive strength of the material [10,11]. However, they are not well-compatible with base epoxy resins [11,12]. Additionally, the disadvantage of phosphorus-containing fire retardants is their lower heat resistance compared to modified epoxy bases [6,7,13,14], which limits their use in engineering plastics and high-temperature binders.

One of the promising classes of compounds that can meet the growing needs of high-tech industries are phosphazenes. The main chain of organophosphazenes consists of alternating phosphorus and nitrogen atoms, and at the phosphorus atom there are organic radicals introduced by substitution of a halogen in halophosphazenes. The nature of organic substituents, usually introduced by the reaction of nucleophilic substitution of chlorine, can vary widely and determines the properties of the final polymer or oligomer.

There are two main synthetic approaches that make it possible to obtain functional phosphazenes capable of forming covalent bonds with epoxy matrices:(1)Synthesis of organophosphazenes with reactive epoxy groups for addition to the epoxy component [15,16,17,18,19,20,21,22,23,24,25,26,27,28,29,30,31,32,33,34,35,36,37,38,39,40].(2)Synthesis of organophosphazenes with reactive amino groups for use as a hardener or its component [41,42,43,44,45,46,47,48,49].

Epoxyphosphazenes are well-compatible with the epoxy matrix and at the same time improve its mechanical properties and heat resistance, probably due to the formation of a special three-dimensional polymer network at the junction nodes, where phosphazene cycles are located [50].

Previously, the synthesis of epoxyphosphazenes was primarily scientific due to the complexity of scaling and the large number of intermediate steps [15,16,17,18,19,21,25,27,28,30,31,32,33]. At present, a number of works have appeared that present fairly simple synthesis methods, with a level of manufacturability close to the production processes of basic epoxy resins [20,22,23,24,26,29,34,37]. Some of them, such as alkoxyphosphazenes, are thermally unstable [51]. Thus, the synthesis of aromatic organophosphazenes is preferred. Examples of such processes are one-stage methods for obtaining: (a) phosphazene-containing epoxy resins based on bisphenol A [52,53] and (b) based on resorcinol [54]. The resulting PhER consisted mainly of tetra- and penta-epoxyphosphazenes and an organic epoxy monomer, which is essentially an active diluent that reduces the viscosity and average functionality to a level acceptable for further processing, and these PhER demonstrated increased mechanical properties and heat resistance. Due to the fact that the viscosity of such resins at ambient temperature is more than 200 Pa∙s, which is a value close to the processing limit, studies were carried out to obtain PhER based on bisphenol A and phenol, including the one-stage method, with the purpose of reducing viscosity and increasing the phosphorus content to reduce flammability [50,55].

In order to obtain PhER with a lower viscosity, which ensures the processability of processing, and a higher phosphorus content, for a potential reduction in flammability, in comparison with PhER described in [52,53,54,55], while maintaining their advantages in mechanical properties, PhER were synthesized by direct interaction of HCP, bisphenol F, and epichlorohydrin. To assess the impact on the physical properties of the use of bisphenol F and the content of the phosphazene fraction, the analysis of the composition of the obtained phosphazene-containing epoxy resins and tests for mechanical properties, glass transition temperature, and viscosity were carried out.

## 2. Materials and Methods

In this article, PhER were synthesized by a single-stage interaction of hexachlorocyclotriphosphazene with an isomer mixture of bisphenol F, in the medium of epichlorohydrin, which acts both as a reagent and as a solvent (Figure 1). The epoxy group formed by the catalyst and the HCl acceptor NaOH was used in one case in solid form, and in the other in the form of an aqueous solution.

### 2.1. Materials

Hexachlorocyclotriphosphazene (HCP), a white crystalline substance (T_m.p._ = 113.0 °C; ^31^P NMR spectrum, singlet with δ_P_ = 19.9 ppm), was obtained by the method in [56].

Bisphenol F (BPF), a white powder (mixture of isomers 2,2′-, 2,4′-, and 4,4′-dioxydiphenyldimethylmethane, T_m.p._ = 162–164 °C) (Manufacturer: Hangzhou Sartort Biopharma Co., Ltd., Hangzhou, China).

Epichlorohydrin (ECH) is a colorless liquid, the content of the main substance is 99.8%, and it was used after preliminary distillation, Т_b.p._ = 116 °С (Manufacturer: Solvay, Tavaux, France).

Potassium hydroxide, in the form of white flakes (the content of the main substance is 90%), was used without purification. The water content was determined by acid-base titration and was about 10% (Manufacturer: JSC “Caustic” Volgograd, Russia).

Acetone is a colorless liquid, and the content of the main substance is 99.8%. It was used as a solvent during isolation without preliminary purification (Supplier: Component-Reaktiv LLC, Moscow, Russia).

Diglycidyl ether of bisphenol A (DGEBA) is a clear-cloudy viscous liquid at room temperature. Commercial-grade KER 828 was used without pre-treatment (Manufacturer: Kumho P & B Chemicals, Inc., Seoul, South Korea).

Diglycidyl ether of bisphenol F (DGEBF) is a clear-cloudy viscous liquid at room temperature. The commercial brand Ipox er 1016 was used without preliminary purification (Supplier: NORTEX LLC, Moscow, Russia).

Diaminodiphenylsulfone (DDS) (Aradur 9664-1) is a fine powder with a particle size smaller than 64 μm and Т_m.p._ = 175 °С (Manufacturer: Huntsman Corporation, Spain).

### 2.2. Synthetic Methods

#### 2.2.1. Synthesis of Phosphazene-Containing Resins Based on Bisphenol F (PNA-BPF) with Batch Introduction of Solid Alkali NaOH

HCP, BPF, and epichlorohydrin were loaded into a 2 L batch reactor equipped with a mechanical stirrer and a direct condenser cooled by recycled water in HCP:BPF ratios from 1:8 to 1:24 with a constant excess of 16-fold ECH (Table 1).

The reaction mixture was heated with stirring to a temperature of 45–50 °C and kept at this temperature until complete dissolution of the solid reagents. Then, the mixture was heated to 77 °C, after which the first portion of sodium hydroxide was loaded in the form of a powder, then at intervals of 5 min, but taking into account that the temperature of the reaction mixture should not exceed 85 °C, the rest of the sodium hydroxide was loaded in equal portions over 1/8 of the total. Sodium hydroxide was taken in a stoichiometric amount. After loading the entire amount of sodium hydroxide, the reaction was carried out for 50 min at a temperature of 80 °C, distilling off the epichlorohydrin–water mixture in a vacuum.

At the end of the reaction, from the resulting suspension, ECH was distilled off on a rotary evaporator, and then the resulting mixture of resin and salt was dissolved in an excess of acetone, and the solution was subsequently filtered from the precipitate through a paper filter, after which acetone was distilled off from the filtrate on a rotary evaporator. The product was a yellow viscous liquid. The yield was 91.2–94.3%.

#### 2.2.2. Synthesis of Phosphazene-Containing Resins Based on Bisphenol F (PNA-BPF-S) with Batch Introduction of NaOH Alkali Solution

The synthesis of PNA-BPF-S was carried out in exactly the same way as the synthesis of PNA-BPF, except that the alkali was loaded in portions in the form of a solution (Table 1). The yield was 76%.

### 2.3. Methods of Analysis

#### 2.3.1. Epoxy Group Content

The epoxy group content was determined by back acid-base titration in acetone according to GOST R 56752-2015 (ISO 3001:1999).

#### 2.3.2. Elemental Analysis

Elemental analysis was carried out by X-ray fluorescence spectrometry with calibration according to the method of fundamental parameters on an X-ray fluorescence spectrometer, ARL PFX-101 (Thermo Fisher Scientific (Ecublens) SARL, Ecublens, Switzerland).

#### 2.3.3. Nuclear Magnetic Resonance Spectroscopy

^31^P NMR spectroscopy was performed on a Bruker AV600 at a working purity of 242 MHz. ^1^H NMR spectroscopy was carried out on a Bruker AV600 instrument at a working purity of 600 MHz. Deuterochloroform was used as a solvent.

#### 2.3.4. Mass Spectrometric Analysis of MALDI-TOF

Mass spectrometric analysis of MALDI-TOF was carried out on a Bruker Auto Flex II instrument.

#### 2.3.5. Rheology

The temperature profile of viscosity was measured on an MCR 302 rheometer (Anton Paar, Graz, Austria). The measurements were carried out in the continuous shear mode with the measuring plate–plate geometry, shear rate of 10 s^−1^, and gap of 1 mm. Top plate diameter was 25 mm.

### 2.4. Method of Curing

The calculated amount of hardener, DDS, was added to the epoxy resin, based on the stoichiometric ratio. The resulting composition was stirred and degassed on an oil bath rotor at 125 °C under a vacuum of 100 mm Hg for 20–40 min until the mixture was completely homogenized. The molds were degreased with acetone before pouring and then treated with an anti-adhesive liquid. All samples were cured at 180 °C for 8 h.

### 2.5. Methods of Testing

#### 2.5.1. Tensile Testing

Tensile tests were carried out on a 50ST (Tinius Olsen, Redhill, United Kingdom) universal electromechanical tensile testing machine according to GOST 11262-2017 (ISO 527-2:2012). Samples of the cured binder were cut from a 4 mm-thick plate on a milling machine in accordance with the paragraph 6.1. standard (Figure 2). The stretching speed was 10 mm/min. From the data obtained, the values of tensile strength, tensile modulus, and elongation were calculated.

#### 2.5.2. Flexural Testing

Flexural tests were carried out on a 50ST electromechanical universal tensile testing machine (Tinius Olsen, Redhill, United Kingdom) according to GOST R 56810-2015 (ASTM D790-10) at a speed of 0.85 mm/min. Samples of the cured binder were cut from a 3.6 mm-thick plate on a milling machine in accordance with the paragraph 6.1. standard. The samples were bars with an aspect ratio of length × width × thickness equal to 80 × 9.8 × 3.6 mm. Span-to-depth ratio was 64:3.6 mm. From the data obtained, the values of flexural strength and flexural modulus were calculated.

#### 2.5.3. Izod Impact Test

Izod impact tests were carried out on an Izod-Charpy K 1053 machine (ATS FAAR, Cassina De’ Pecchi, Italy) according to GOST 19109-2017 (ISO 180:2000), “Plastics. Izod impact strength method”, which is a modified ISO 180:2000 standard. The samples were bars with an aspect ratio of length × width × thickness equal to 80 × 9.8 × 3.6 mm.

#### 2.5.4. Glass Transition Temperature

The glass transition temperature was determined on a DSC 204 F1 Phoenix differential scanning calorimeter (Netzsch, Selb, Germany) according to GOST R 55135-2012 (ISO 11357-2:1999). Under the dynamic heating mode, the temperature range was 40–270 °C, and the heating rate was 10 °C/min. All tests were carried out under a nitrogen atmosphere with a flow rate of 60–100 mL/min. The weight of the samples was 5–10 mg.

The device was calibrated for temperature and enthalpy for standard metals gallium, indium, tin, bismuth, and zinc.

The analysis of parameters and characteristic points was carried out in the Proteus thermal analysis software.

## 3. Results and Discussion

### 3.1. Synthesis and Study of PNA-BPF and PNA-BPF-S

Analysis of the composition and structure of the obtained PhER was carried out using MALDI-TOF spectrometry and ^31^P NMR spectroscopy.

MALDI-TOF analysis showed the presence in the products of compounds with *m*/*z* values and possible structures, as presented in Table 2.

The formation of VII-membered spirocyclic fragments at the phosphorus atom with the participation of the 2,2-isomer of bisphenol F in almost every component of the product mixture is noteworthy. This leads to a decrease in the average functionality of the mixture in comparison with epoxyphosphazenes based on bisphenol A and resorcinol, which can favorably affect the pot life of the resins.

MALDI-TOF analysis is not an unambiguously quantitative method of analysis without confirmation of its results by other methods, however it can be used to determine the qualitative presence of compounds and trends in their formation depending on the conditions. According to the results of MALDI-TOF, presented in Table 3 and Table 4 and Figure 3 and Figure 4, there are pronounced extreme dependencies of the formation of compounds with structures I and V with peaks at a ratio of 1:12. A less pronounced increase in the formation of compounds VI, VIII, and X, and a decline for compounds III, VII, and IX, were observed with peaks at a ratio of 1:18. The amounts of compounds II and IV fluctuate without unambiguous dependencies. Extreme values of the formation of substances are typical for synthesis 4 with a ratio of 1:18.

^31^P NMR spectra of the final product (Figure 5) indicate incomplete substitution and the formation of a mixture of tetra-, penta-, and hexa-derivatives of hexachlorocyclo-triphosphazene with a predominance of penta-derivatives. Due to the presence of singlet signals in the region of 7.07, 7.51, and 8.08 ppm, it can be assumed that several types of hexa-substituted triphosphazenes are formed (Table 4). However, due to steric difficulties, the formation of compounds XIV, XV, and XVI is unlikely; therefore, the three signals mentioned above probably correspond to compounds XI, XII, and XIII. The presence of compound XIII is also confirmed by MALDI-TOF analysis and corresponds to compound VI. As the BPF excess increases, the yield of hexa-substituted derivatives with a singlet signal of 8.08 ppm decreases, and the yield of hexa-substituted derivatives of the structure increases with signals at 7.51 and 7.07. The penta-derivative penta I corresponds to the doublet (4.88–5.21 ppm) and triplet (20.08–20.75 ppm) systems. The penta-derivative penta II corresponds to the doublet (5.64–5.99 ppm) and triplet (20.75–21.48 ppm) systems. In the penta-derivatives penta I and penta II, the triplet peaks in the region of 20.78 ppm overlap, which is confirmed by the values of the integral areas—the integral area under the peak with overlap is equal to the sum of the areas under the peaks in the region at 21.48 and 20.08 ppm. A tetrasubstituted product corresponds to a triplet (2.96 ppm) and a doublet (19.41–19.74 ppm) signal system. Derivatives of hexa I, II, III, and penta I and II may not be individual substances, but a mixture of derivatives with the same degree of substitution, but different substituents—epoxy, spirocyclic, and bridging—which produce signals in one or close intervals, which can lead to their overlap.

^1^H NMR spectra of the obtained epoxy oligomers (Figure 6) are similar to the spectra for dianic epoxy resins (industrial grades ED-20, KER-828, and their analogues); however, the signals in the region of 3.55–3.95 ppm indicate the presence of a small amount of glycol end groups formed as a result of the hydrolysis of epoxy groups, apparently caused by the presence of residual alkali in the reaction medium, which was not completely consumed due to incomplete substitution of chlorine atoms in hexachlorocyclotriphosphazene. The results of the integration of signal systems are consistent with theoretical calculations.

Table 5 shows the PhER yields depending on the ratios of the initial reagents. The values of the yields of syntheses 1–5 are close and are associated with losses during filtration from salts during the isolation process. The lower yield in the synthesis of PNA-BPF-S-1 is associated with the formation of sparingly soluble oligomers, which were deposited on the filter along with the salt during filtration, which also affected the deviation in the epoxy number.

Table 6 presents the technological characteristics of various base and phosphazene-containing epoxy resins. Among all pairs of base and phosphazene-containing epoxy resins based on the same diphenols, there is a pattern of increase in viscosity and decrease in the epoxy number in PhER. An exception can be considered PhER obtained by the interaction of HCP with BPA and phenol. By using phenol, this type of resin reduces the viscosity and increases the phosphorus content, which can improve the fire resistance. However, at the same time, the epoxy number drops significantly for these resins, which will reduce the crosslink density during curing, and may worsen the mechanical properties. The described pattern is observed in the case of PhER based on BPF; first, when epoxyphosphazenes appear in the composition, the mechanical properties increase compared to the base resin, then, with an increase in the proportion of the phosphazene fraction and a decrease in the epoxy number, the mechanical properties either remain at the same level, or fall (Table 7 and Table 8).

Figure 7 shows the temperature dependence of the viscosity of PhER compared to base epoxy resins. With a decrease in the higher molecular weight phosphazene component and a change in the substituent from bisphenol A to bisphenol F, when receiving PhER (Table 6), the viscosity also naturally decreases.

### 3.2. Testing of Cured Compositions PNA-BPF and PNA-BPF-S

With an increase in the degree of substitution along with an increase in the excess of bisphenol, which is described by the interpretation of ^31^P NMR spectra (Figure 5), and the overall increase in the content of phosphazene, the dynamics of changes in the physical and mechanical properties of cured epoxy resins with diaminodiphenyl sulfone at 180 °C for 8 h are also complexly correlated. Extreme dependencies of the formation of a number of compounds and fluctuations in their content, which is described in Section 3.1 when interpreting the MALDI-TOF spectra, could affect the fluctuations in the values of mechanical properties (Table 7 and Table 8).

With an increase in the phosphazene fraction, first, an increase in the strength and elasticity of the cured PhER during tensile testing was observed, then a tendency to a decrease in tensile modulus (Figure 8). In flexural tests, an increase in strength was also observed, while the trends in elasticity and flexural modulus were different to tensile tests: the flexural elongation at break continuously decreased with increasing content of the phosphazene fraction, while flexural modulus remained unchanged (Figure 9).

Similar dependencies were also observed when testing for impact strength (Figure 10). With a decrease in the phosphazene fraction, the impact resistance increased to its maximum in the synthesis of PNA-BPF-5, but at the same time it decreased again for the DGEBF epoxy resin, which does not contain epoxyphosphazenes. Based on this, one can talk about the existing modifying property of epoxyphosphazenes based on bisphenol F with respect to the base DGEBF resin. The modifying effect is most likely maximal at the content of the phosphazene fraction in the range of 10–25%. A similar dependence was observed earlier in the study of the properties of PhER based on bisphenol A in [58].

Appendix A show graphs of percentage changes in the mechanical properties of neat and phosphazene-containing resins based on bisphenol F relative to DGEBA.

A different kind of dependence was observed for the glass transition temperature. Its values decreased compared to the DGEBF neat resin (Table 8), with a minimum content of the phosphazene fraction in PNA-BPF-5, after which they passed through a maximum in PNA-BPF-3 with indicators at the level of base epoxy resins, and then with an increased PhER content, continued to drop significantly, which was not observed for PhER of other types [58].

It is likely that with a further decrease in the content of the phosphazene fraction, the glass transition temperature will resume growth, which may coincide with the maxima of the values of impact strength, elasticity in tension and flexion, and viscosity, while maintaining or some increase in the tensile and flexural properties, according to comparison with the base resin DGEBF, which was observed for all experiments during synthesis on the solid alkali.

Despite the sharply different composition of PNA-BPF-4, corresponding to a phosphazene content of 28%, this was not reflected in the physical and mechanical properties. A sharp change in properties was observed immediately at the lowest phosphazene content of 22%, then the properties either remained at the same level or a tendency to decrease was observed. This may be due to a double modifying effect of the phosphazene content: factor 1 is an increase in the total phosphazene content, and factor 2 is a change in the ratio of the various epoxyphosphazenes (compounds I–X) formed depending on the proportions of the initial reagents. Comprehensively, we can speak of an important effect on the improvement of the properties of compounds II and VII, which constitute the main part for all syntheses. For the remaining compounds, more pronounced fluctuations and extreme dependencies in content were observed with an increase in the proportion of phosphazene, which could affect the general trend of a gradual or significant decrease in properties with an increase in the proportion of phosphazene after an initial sharp increase. The difficulty of assessing factor 2 lies in the need to obtain individual substances formed during the synthesis, and for this it is necessary to develop a method for separating the resulting mixture of epoxyphosphazene resins into individual substances, after which it will be possible to evaluate the modifying effect of the content of each epoxyphosphazene. This paper presents a comprehensive assessment of the effect of the phosphazene content in the production of PhER by a one-stage method, which may be of interest from the perspective of commercial production. Based on the results obtained during the tests, to find the best combination of properties, it is worth studying PhER with a content of 10–25% obtained by this method.

## 4. Conclusions

The phosphazene-containing epoxy resins obtained in this work have the content of epoxy groups of 15.2–22.8%, and phosphorus of 1.5–3.9%. With a decrease in the content of HCP in the initial reagents, an increase in the mechanical properties of the cured resulting epoxy resins was observed. The most successful in terms of their performance were samples with a ratio of initial reagents HCP:BPF:ECH of 1:12:16–1:24:16. They demonstrated an increase in impact strength by 40–80%, relative elongation at fracture in tension and in flexion by 50–100%, while maintaining glass transition temperature and tensile and flexural strength compared to base epoxy resins based on bisphenols A and F. These PhER can be cured with conventional hardeners to obtain materials with mechanical properties and heat resistance above the base commercial grades of epoxy resins and potentially lower combustibility due to the content of phosphorus.

Thus, phosphazene-containing epoxy resins based on bisphenol F, with a low content of the phosphazene fraction, can act as modifiers for traditional epoxy resins, being compatible with them, to increase impact resistance and elasticity while maintaining other basic mechanical and technological characteristics, and can be used as a binder component for composite materials, adhesives, and paints.

## Figures and Tables

**Figure 1 polymers-14-04547-f001:**
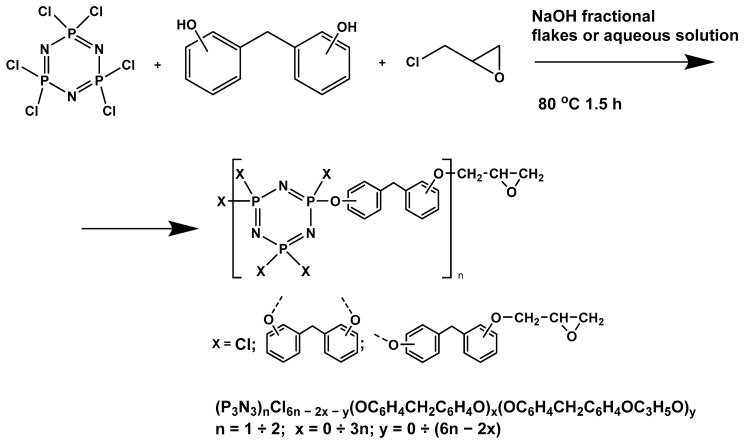
Single-stage synthesis of phosphazene-containing epoxy resins by direct interaction of hexachlorocyclotriphosphazene (HCP) and bisphenol F in epichlorohydrin medium (this work).

**Figure 2 polymers-14-04547-f002:**
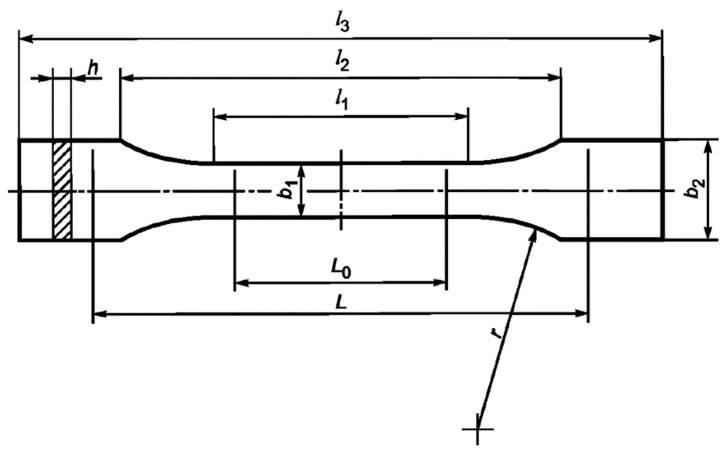
Tensile test sample. Dimensions, in mm: l1 = 80 ± 2, l2 = 109.3 ± 3.2, l3 = 170, L0 = 75 ± 0.5, L = 115 ± 1, b1 = 10 ± 0.2, b2 = 20 ± 0.2, h = 4 ± 0.2, r = 24 ± 1.

**Figure 3 polymers-14-04547-f003:**
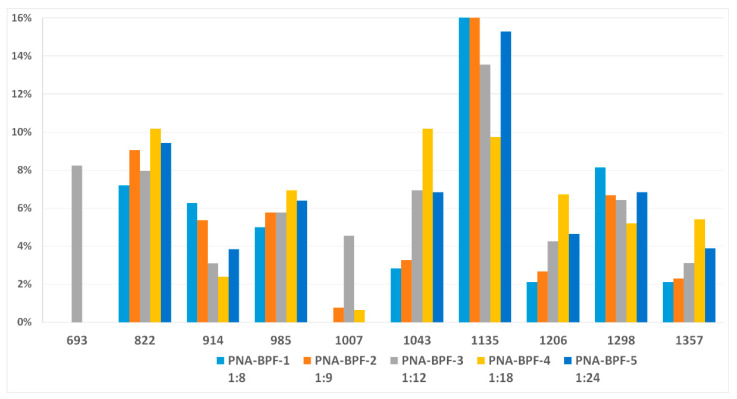
The content of compounds I–X (Table 2) in samples PNA-BPF-1 ÷ 5 ((1)–(5), respectively).

**Figure 4 polymers-14-04547-f004:**
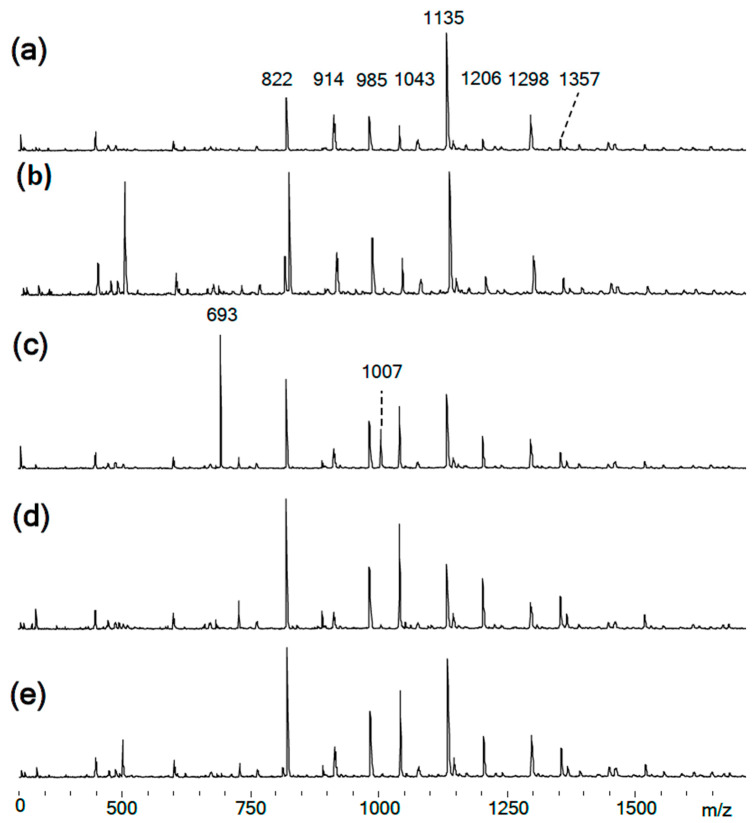
(**a–e**) MALDI-TOF spectra of the obtained PhER syntheses of PNA-BPF-1 ÷ 5, respectively.

**Figure 5 polymers-14-04547-f005:**
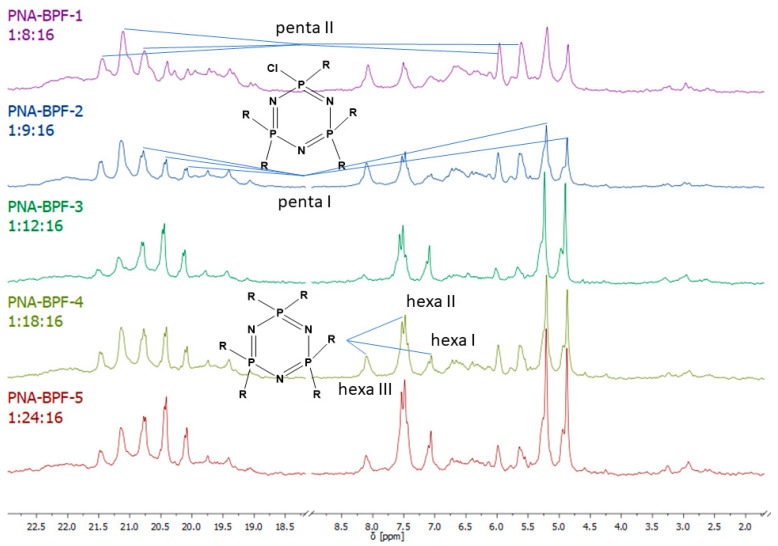
^31^P NMR spectra of PNA-BPF.

**Figure 6 polymers-14-04547-f006:**
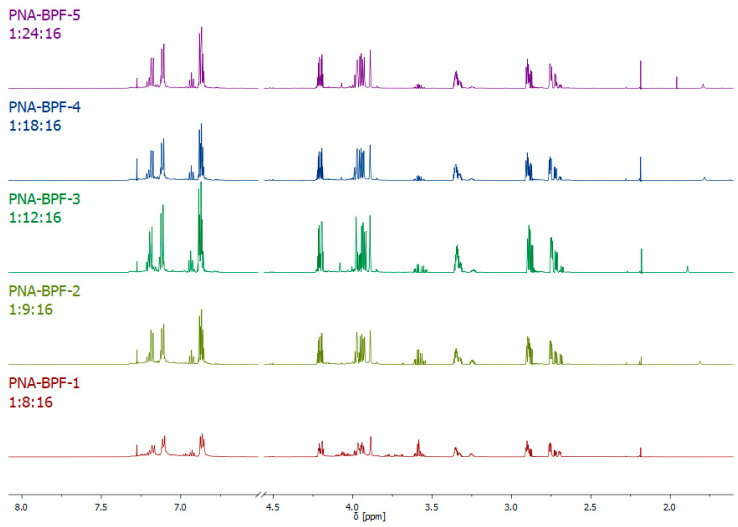
^1^H NMR spectra of PNA-BPF.

**Figure 7 polymers-14-04547-f007:**
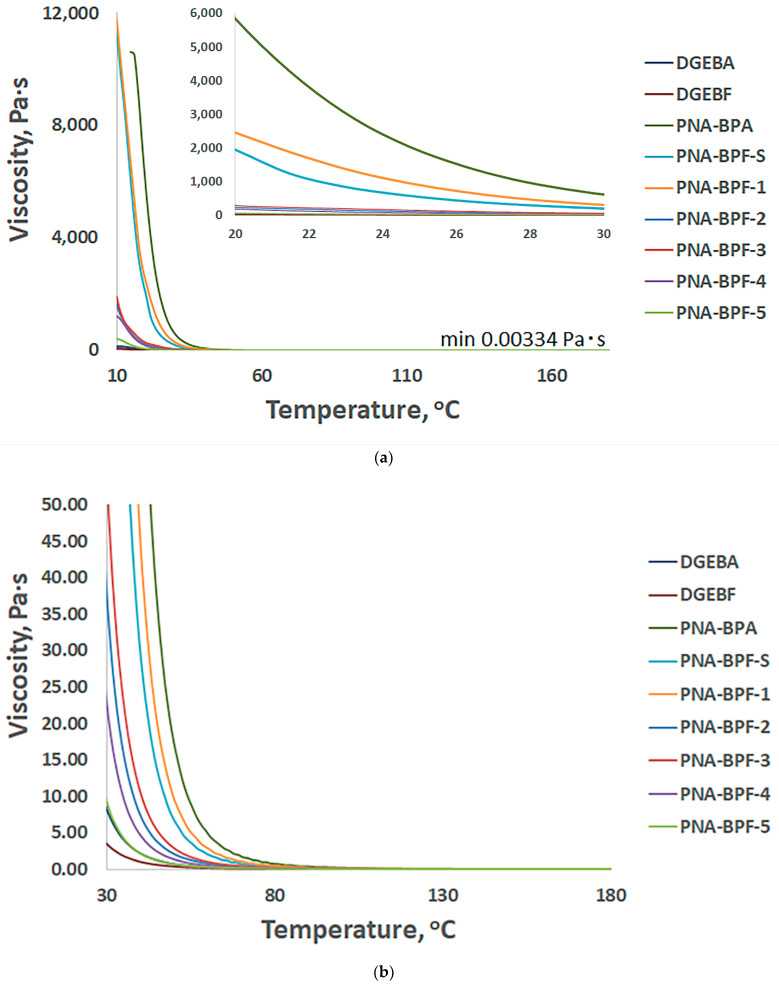
Temperature profile of viscosity versus temperature of obtained PhER, (**a**) from 10 to 180 °C and from 22 to 26 °C, and (**b**) from 30 to 170 °C, respectively. PNA-BPA—epoxy resin obtained by the method in [52] at a ratio of HCP:BPA equal to 1:9.

**Figure 8 polymers-14-04547-f008:**
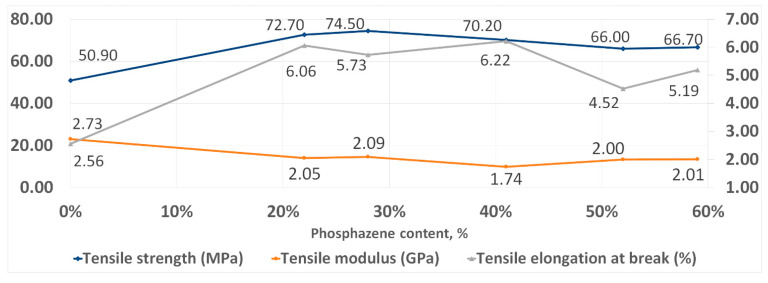
Tensile strength properties of cured resins.

**Figure 9 polymers-14-04547-f009:**
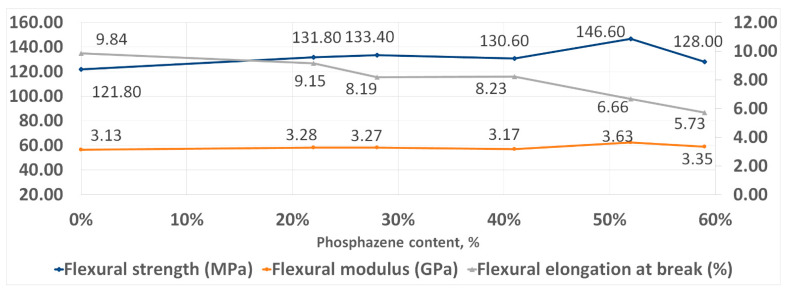
Flexural strength properties of cured resins.

**Figure 10 polymers-14-04547-f010:**
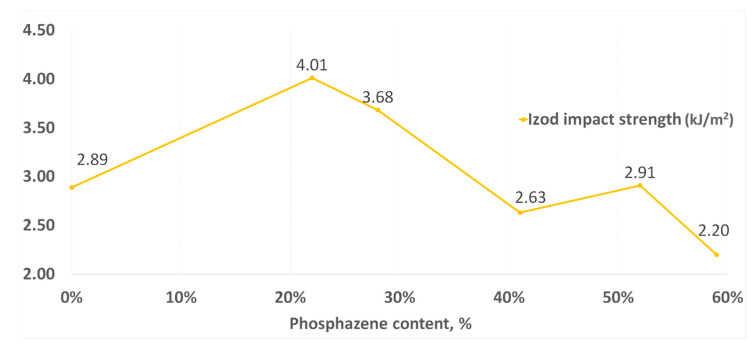
Impact strength according to Izod.

**Table 1 polymers-14-04547-t001:** The amounts of initial reagents for the synthesis of PhER.

Sample	Molar Ratio HCP:BPF:ECH	HCPg (Mole)	BPFg (Mole)	ECHmL (Mole)	NaOHg (Mole)
**PNA-BPF-S**	1:8:16	35 (0.101)	161.11 (0.805)	1387.865 (17.606)	64.363 (1.609)
**PNA-BPF-1**	1:8:16	35 (0.101)	161.11 (0.805)	1387.865 (17.606)	64.363 (1.609)
**PNA-BPF-2**	1:9:16	35 (0.086)	181.249 (0.905)	1639.022 (20.917)	72.408 (1.810)
**PNA-BPF-3**	1:12:16	23 (0.066)	158.809 (0.793)	1574.182 (20.096)	63.444 (1.586)
**PNA-BPF-4**	1:18:16	15 (0.043)	155.356 (0.776)	1675.045 (21.376)	62.064 (1.552)
**PNA-BPF-5**	1:24:16	10.5 (0.030)	144.999 (0.724)	1626.414 (20.757)	57.927 (1.448)

**Table 2 polymers-14-04547-t002:** Possible structures and formulas of the main compounds formed during the synthesis of PNA-BPF and PNA-BPF-S.

*m*/*z*	Structure and Formula ^1^	No. of Structure
693	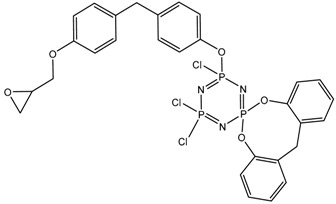 P_3_N_3_Cl_3_(OArO)(OArOGly)	I
822	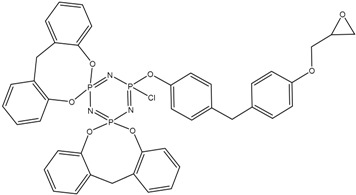 P_3_N_3_Cl(OArO)_2_(OArOGly)	II
914	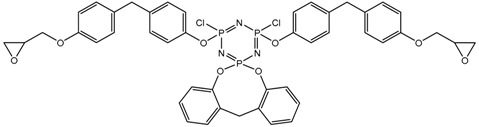 P_3_N_3_Cl_2_(OArO)(OArOGly)_2_	III
985	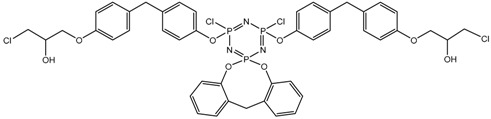 P_3_N_3_Cl_2_(OArO)(OArOGly’)_2_	IV
1007	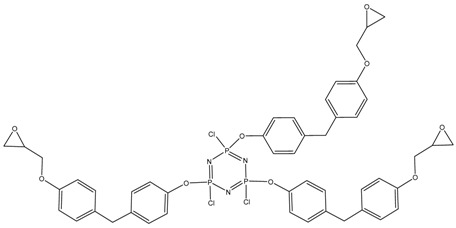 P_3_N_3_Cl_3_(OArOGly)_3_	V
1043	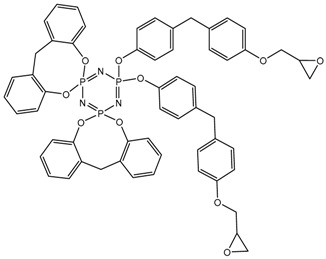 P_3_N_3_(OArO)_2_(OArOGly)_2_	VI
1135	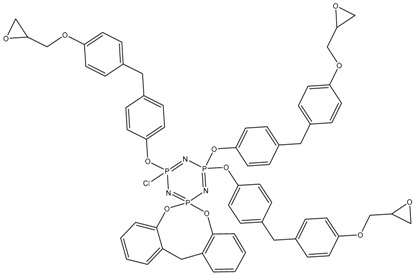 P_3_N_3_Cl(OArO)(OArOGly)_3_	VII
1206	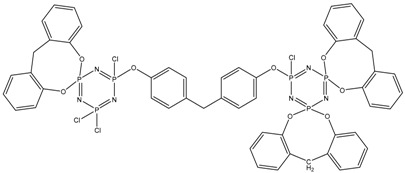 (P_3_N_3_)_2_Cl_4_(OArO)_4_	VIII
1298	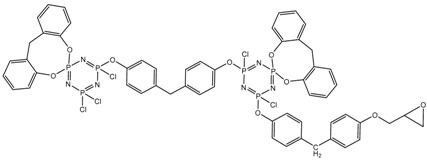 (P_3_N_3_)_2_Cl_5_(OArO)_3_(OArOGly)	IX
1357	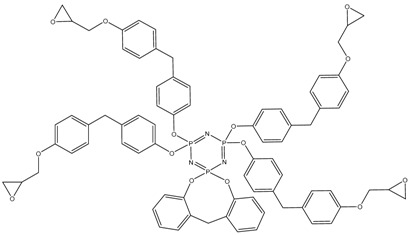 P_3_N_3_(OArO) (OArOGly)_4_	X

^1^ AR = 
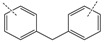
, Gly = 
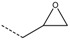
, Gly’ = 
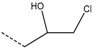
.

**Table 3 polymers-14-04547-t003:** The main compounds formed during the synthesis of PNA-BPF and PNA-BPF-S, the share of which is more than 4% in at least one of the experiments.

*m*/*z*	Formula	Content (wt.%)				
		PNA-BPF-1	PNA-BPF-2	PNA-BPF-3	PNA-BPF-4	PNA-BPF-5
693	P_3_N_3_Cl_3_(OArO)(OArOGly) (I)	0	0	8	0	0
822	P_3_N_3_Cl(OArO)_2_(OArOGly) (II)	7	9	8	10	9
914	P_3_N_3_Cl_2_(OArO)(OArOGly)_2_ (III)	6	5	3	2	4
985	P_3_N_3_Cl_2_(OArO)(OArOGly’)_2_ (IV)	5	6	6	7	6
1007	P_3_N_3_Cl_3_(OArOGly)_3_ (V)	0	1	5	1	0
1043	P_3_N_3_(OArO)_2_(OArOGly)_2_ (VI)	3	3	7	10	7
1135	P_3_N_3_Cl(OArO)(OArOGly)_3_ (VII)	21	16	14	10	15
1206	(P_3_N_3_)_2_Cl_4_(OArO)_4_ (VIII)	2	3	4	7	5
1298	(P_3_N_3_)_2_Cl_5_(OArO)_3_(OArOGly) (IX)	8	7	6	5	7
1357	P_3_N_3_(OArO)(OArOGly)_4_ (X)	2	2	3	5	4
Total proportion of major compounds		54	52	64	57	57

**Table 4 polymers-14-04547-t004:** Possible structures and formulas of hexa-substituted phosphazenes formed during the synthesis of PNA-BPF.

Structure and Formula	No. of Structure
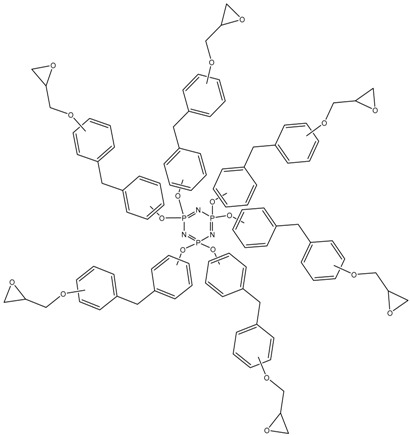	XI
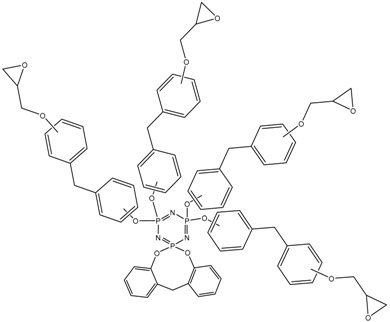	XII
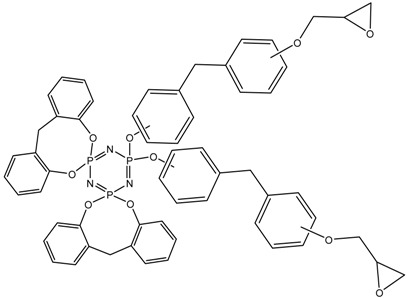	XIII
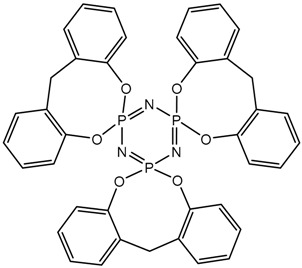	XIV
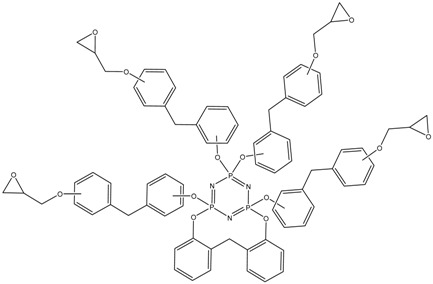	XV
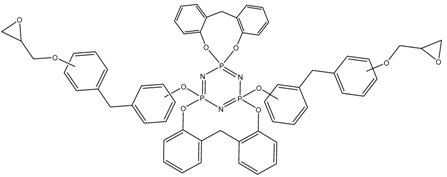	XVI

**Table 5 polymers-14-04547-t005:** Yield of the obtained PhER.

Sample	Molar Ratio HCP:BPF:ECH	Yield	
		g	%
PNA-BPF-S-1	1:8:16	180	76.0
PNA-BPF-1	1:8:16	216	91.2
PNA-BPF-2	1:9:16	247	92.2
PNA-BPF-3	1:12:16	223	94.0
PNA-BPF-4	1:18:16	220	93.5
PNA-BPF-5	1:24:16	209	94.3

**Table 6 polymers-14-04547-t006:** Comparison of technological characteristics of base and phosphazene-containing epoxy resins based on basic diphenols and the content of phosphorus and chlorine between different PhER. The content of P and Cl wt.% in PhER obtained in this work, determined by X-ray fluorescence spectrometry.

Raw Reagents Ratio		Mixture Average Functionality	Content (wt.%)				Viscosity (Pa∙s) at the Temperature of (°С)		
			Epoxy Group (%)	P	Cl	Phosphazene Fraction ^1^	20	40	70
		DGEBA							
		2.0	22.8	-	-	-	41.72	2.17	0.15
		DGEBF							
		2.0	24.6–27.0	-	-	-	16.31	0.92	0.09
		RDGE							
		2.0	34.4–36.4	-	-	-	1.10	0.11	0.03
**HCP:BPA**		PhER obtained by interaction of HCP with BPA and ECH [52,53,57]							
1:8		2.5	17.1	3.1	2.7	49	-	220	3
1:9 ^2^		2.5	18.1	3.3	2.4	52	5855	85	2
1:12		2.3	20.0	1.8	1.5	30	-	130	2
1:16		2.2	21.4	1.5	1.3	25	440	78	2
**HCP:Resorcinol**		PhER obtained by interaction of HCP with resorcinol and ECH [54]							
1:12		2.4	21.0	4.0	4.4	43	8.33	6.15	0.36
1:16		2.3	28.6	3.0	2.4	32	2.43	1.94	0.15
1:24		2.2	29.6	2.0	1.9	21	1.71	0.45	0.05
**HCP:PhOH:BPA**		PhER obtained by interaction of HCP with BPA, phenol, and ECH [55]							
1:2:6		2.2	16.1	4.6	2.2	54	64.6	13.7	0.8
1:3:5		2.0	15.5	5.0	2.3	51	58.6	10.6	0.8
1:4:4		1.9	14.7	5.4	2.7	47	9.4	6.0	0.8
**HCP:BPF:ECH**		PhER obtained by interaction of HCP with BPF and ECH (this work)							
1:8:16 (NaOH solution)		2.8	15.2	3.7	4.2	56	1942.00	38.68	0.73
1:8:16		2.8	18.1	3.9	2.1	59	2463.11	45.03	1.04
1:9:16		2.7	20.5	3.4	1.9	52	229.24	7.45	0.34
1:12:16		2.4	20.4	2.7	1.6	41	282.99	10.65	0.43
1:18:16		2.2	22.8	1.8	1.4	28	181.12	4.69	0.22
1:24:16		2.2	21.5	1.5	1.3	22	62.65	2.16	0.13

^1^ Values for this work are calculated based on the phosphorus content. ^2^ Synthesis at this ratio was performed within the framework of this work.

**Table 7 polymers-14-04547-t007:** Tensile and impact properties and glass transition temperature of the cured epoxy resins compared to base epoxy resins.

Sample (HCP:BPF:ECH)	Tensile Strength (MPa)	Tensile Modulus (GPa)	Tensile Elongation at Break (%)	Izod Impact Strength (kJ/m^2^)	Tg (°C)
DGEBA	69.8	2.6	4.1	2.1	169.2
DGEBF	50.9	2.7	2.6	2.9	174.8
PNA-BPF-5 (1:24:16)	72.7	2.1	6.1	4.1	156.7
PNA-BPF-4 (1:18:16)	74.5	2.1	5.7	3.7	169.3
PNA-BPF-3 (1:12:16)	70.2	1.7	6.2	2.6	174.7
PNA-BPF-2 (1:9:16)	66.0	2.0	4.5	2.9	153.0
PNA-BPF-1 (1:8:16)	66.7	2.0	5.2	2.2	160.0
PNA-BPF-S (1:8:16)	29.4	2.6	1.2	1.4	141.8

**Table 8 polymers-14-04547-t008:** Flexural properties of the cured epoxy resins compared to base epoxy resins.

Sample (HCP:BPF:ECH)	Flexural Strength (MPa)	Flexural Modulus (GPa)	Flexural Elongation at Break (%)
DGEBA	125.6	3.02	6.99
DGEBF	121.8	3.13	9.84
PNA-BPF-5 (1:24:16)	131.8	3.28	9.15
PNA-BPF-4 (1:18:16)	133.4	3.27	8.19
PNA-BPF-3 (1:12:16)	130.6	3.17	8.23
PNA-BPF-2 (1:9:16)	146.6	3.63	6.66
PNA-BPF-1 (1:8:16)	128.0	3.35	5.73
PNA-BPF-S (1:8:16)	108.3	4.28	2.68

## Data Availability

The corresponding author can be approached for the availability of data.

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
