# Peer review of "Phosphazene-Containing Epoxy Resins Based on Bisphenol F with Enhanced Heat Resistance and Mechanical Properties: Synthesis and Properties"

_polymers, 2022, doi:10.3390/polym14214547_

Round 1
Reviewer 1 Report
This manuscript presents mainly the thermal, rheological and mechanical properties of phosphazene-containing epoxy resins based on bisphenol F with varying phosphazene contents. The idea of this manuscript is interesting and is believed to be the interest to the scientific community. However, some weaknesses need to be addressed before being accepted for publication. The comments are shown as follows.
1. Please seek an English language editor to improve the language quality. Many incomplete sentences have been noticed throughout the manuscript. Furthermore, certain parts of this work are difficult to follow. The level of expression should be substantially enhanced.
2. The abstract should summarise the entire study very precisely. It should contain background, objectives, experimental method, major findings and conclusion. However, it seems like some elements are not included in the abstract.
3. On page 1, please rewrite the third paragraph of the introduction, as this sentence is confusing and not understandable. Are low and unstable user characteristics considered their advantage? Moreover, this paragraph can be combined with the subsequent paragraph as it has only one sentence.
4. There are many paragraphs consisting of only one or a few sentences. Please combine them with the subsequent paragraphs.
5. Novelty/innovative aspects of this work should be stressed in the last paragraph of the introduction.
6. On page 3, the acronym for Bisphenol F is ‘BPhF’. However, the authors have used a different acronym, ‘BPF’, for the Bisphenol F throughout the manuscript. Please check.
7. On page 4; the last sentence, what does 91.2 – 94.3 % represent? Further elaboration is needed to improve its clarity.
8. On page 5; sub-sections 2.5.1 and 2.5.2, what does paragraph 6.1 standard mean?
9. On page 5 and 6, please provide the dimensions of the specimens for each mechanical test. In addition, please also provide the span-to-depth ratio for the flexural test.
10. On page 5, please change ‘universal electromechanical tensile testing machine’ to ‘electromechanical universal testing machine’.
11. On page 6, please change ‘Izod impact strength test’ to ‘Izod impact test’. In addition, please also provide the type and model of the machine for the Izod impact tests.
12. On page 13, Table 5 and Table 6 are not mentioned and explained in the text.
13. On page 16, what does the dynamic of changes mean? Further explanation and clarification are needed.
14. On page 16, the last paragraph is not appropriate. It does not exactly reflect the results recorded in Table 9 and Figure 8. The flexural strength actually dropped with a decrease in the phosphazene content. In particular, it does not significantly impact the flexural strength of the materials, as it can be seen that the flexural strengths of those specimens are similar except for those with 50 % phosphazene.
15. On pages 17 and 18, what do the relative values represent? I don’t think these plots (Figure 7(b), Figure 8(b) and Figure 9(b)) are crucial.
16. On page 18, avoid using first-person pronouns in a research article.
17. Overall, the knowledge depth of the entire discussion is insufficient. It seems like the authors have only analysed the results without sufficient discussion and evidence. The authors need to explain the trend of the mechanical properties and the role of phosphazene in affecting the mechanical properties of the materials.

Reviewer 2 Report
Dear Authors
This manuscript studies Phosphazene-Containing Epoxy Resins Based on Bisphenol F with Enhanced Heat Resistance and Mechanical Properties: Synthesis and Properties. The experiment and measurement techniques conducted properly especially FT-IR analysis and 31P NMR. I recommend to accept publishing this manuscript in Polymers.
best regards
